# Mechanisms of Immune Checkpoint Inhibitor Resistance in Hepatocellular Carcinoma and Strategies for Reversal

**DOI:** 10.3390/cancers18010039

**Published:** 2025-12-22

**Authors:** Xin-Ye Dai, Xiao-Juan Yang, Hong Wu, Ying-Hao Lv, Tian Lan

**Affiliations:** 1Department of General Surgery, West China Hospital, Sichuan University, Chengdu 610041, China; daixinye@stu.scu.edu.cn (X.-Y.D.); wuhong@scu.edu.cn (H.W.); 2Liver Transplant Center, Transplant Center, West China Hospital, Sichuan University, Chengdu 610041, China; 3Laboratory of Hepatic AI Translation, Frontiers Science Center for Disease-Related Molecular Network, West China Hospital, Sichuan University, Chengdu 610041, China; yxiaojuan@scu.edu.cn

**Keywords:** hepatocellular carcinoma, immune checkpoint blockade, immune therapeutic resistance, reversal strategies

## Abstract

Immune checkpoint inhibitors (ICIs) have become an important treatment for hepatocellular carcinoma (HCC), a common form of liver cancer. However, a significant number of patients do not respond well to this treatment, and many who initially benefit eventually develop resistance. This research aims to better understand the mechanisms behind resistance to ICIs in HCC, which could help improve patient outcomes. By identifying factors such as immune system dysfunction, changes in the tumor microenvironment, and genetic mutations, the study seeks to develop strategies to overcome these barriers. The findings may guide future treatment approaches, combining ICIs with other therapies to enhance effectiveness, offering hope for more patients with this challenging disease.

## 1. Introduction

Hepatocellular carcinoma (HCC) is the most prevalent malignant tumor of the liver. In recent years, HCC has emerged as the sixth most common malignancy and the third leading cause of cancer-related mortality worldwide [1]. The treatment of HCC is typically stratified based on the stage of the disease, liver function, and the presence of underlying cirrhosis. For early-stage HCC, surgical options such as liver transplantation, partial hepatectomy, or ablation therapies (e.g., radiofrequency ablation (RFA), transcatheter arterial chemoembolization (TACE)) are preferred [2]. Approximately 50–60% of patients will require systemic therapy due to the frequent presentation of patients with advanced-stage disease at diagnosis and the high rate of postoperative recurrence [3,4]. The commonly used systemic therapeutic drugs can be classified into two major categories: targeted drugs and immune checkpoint inhibitors (ICIs). Targeted drugs include tyrosine kinase inhibitors (TKIs) such as sorafenib and lenvatinib, as well as VEGF inhibitors such as bevacizumab and ramucirumab. ICIs include PD-1 and PD-L1 inhibitors such as atezolizumab and pembrolizumab, as well as CTLA-4 inhibitors such as ipilimumab and tremelimumab. Previously, sorafenib was the only systemic treatment option available for patients with advanced HCC. Recently, ICIs and targeted-immunotherapy combination therapy have been widely used in the treatment of HCC. Atezolizumab combined with bevacizumab has become the preferred first-line systemic treatment option for advanced HCC [2]. IMbrave150 showed that the objective response rate (ORR) of atezolizumab combined with bevacizumab was 27.3%, while that of sorafenib was 11.9% [5]. In addition, the trial demonstrated significant improvements in both progression-free survival (PFS; 6.8 vs. 4.3 months) and overall survival (OS; 19.2 vs. 13.2 months), establishing this regimen as a superior first-line option. Another first-line systemic treatment option is tremelimumab combined with durvalumab. The results of the phase III HIMALAYA trial showed that the ORR of the combined treatment group reached 20.1%. Furthermore, the HIMALAYA regimen improved OS (16.4 months vs. 13.8 months with sorafenib), while maintaining a comparable PFS (3.8 months) [6]. However, the response rate of patients to single-agent PD-1 blockade is between approximately 15% and 20% [7,8,9]. In the phase II clinical trial of KEYNOTE 224, 51 patients with untreated HCC were treated with the anti-PD-1 antibody pembrolizumab. The reported median ORR was 16% (95% confidence interval: 7% to 29%). Although KEYNOTE 224 showed durable responses in some patients, the median PFS remained modest (4 months), underscoring the limitations of PD-1 monotherapy [10]. The clinical efficacy of immunotherapy remains limited. There is an urgent need to improve the stratification of patients who may benefit from these treatment options and to reveal the different mechanisms by which cancer cells promote resistance to immune-based therapies in HCC. This will address the issue of developing predictive markers and pave the way for improving combined strategies.

This article systematically reviews the potential mechanisms by which ICIs induce primary and acquired resistance in HCC, covering aspects such as abnormal regulation of the tumor immune microenvironment (TIME), exhaustion of T cell function, and abnormal activation of pathways within tumor cells. Based on this, it further explores current promising combined treatment strategies aimed at overcoming ICI resistance, providing a theoretical basis and research directions for future precision immunotherapy of HCC and improving the long-term prognosis of patients.

## 2. Mechanisms of ICI Resistance

Resistance to ICIs can be classified into primary and acquired resistance. Primary resistance refers to patients who exhibit no response to ICIs from the onset of treatment, resulting in rapid or eventual disease progression. In contrast, acquired resistance occurs in patients who initially respond to ICIs but subsequently experience clinical and/or radiological disease progression [11,12]. Table 1 summarizes the mechanisms of ICI resistance.

### 2.1. Primary Resistance

As shown in Figure 1, the intracellular mechanisms include the absence of neoantigens, defects in MHC-I, abnormal tumor cell signaling pathways, tumor-associated exosomes, and metabolic reprogramming, while the extracellular mechanisms include T cell dysfunction and the immunosuppressive cells in the microenvironment.

#### 2.1.1. Absence of Neoantigens

Genetic mutations can lead to the generation of immunogenic neoantigens [13]. In other cancers, a high tumor mutational burden (TMB) accompanied by increased expression of neoantigens is associated with a favorable response to ICI therapy [14]. In contrast, HCC typically exhibits a low-to-moderate TMB [3], with an average of 5 individual cell mutations per megabase, corresponding to approximately 60 non-synonymous mutations [15]. This may contribute to the primary resistance of HCC to immunotherapy.

#### 2.1.2. Defects in MHC-I

T cell-mediated immune activation relies on the specific recognition of antigens presented by the major histocompatibility complex (MHC) on the surface of antigen-presenting cells (APCs) [16]. The downregulation or absence of MHC class I (MHC-I) molecules in HCC cells restricts the presentation of tumor antigens to CD8+ T cells, thereby preventing the immune system from effectively recognizing tumor cells [17,18]. In HCC, the levels of MHC-I molecules vary across different cell lines, with high MHC-I expression being associated with increased CD3+ T cell infiltration and a favorable prognosis [19]. Defects in antigen-presenting molecules, such as low-molecular-weight protein 2, low-molecular-weight protein 7, and TAP1, may lead to the downregulation of MHC-I expression levels and impaired antigen presentation, enabling tumor cells to evade the cytotoxic effects of specific cytotoxic T lymphocytes (CTLs) [20]. In HCC, the epidermal growth factor receptor (EGFR) is often overexpressed. Both in vivo and in vitro studies have shown that EGFR signaling can downregulate the expression level of MHC-I, and this effect can be reversed by the EGFR inhibitor gefitinib [21].

#### 2.1.3. Signal Pathway

Next-generation sequencing (NGS) analysis of tumor tissue from HCC patients treated with ICIs revealed that aberrations in the Wnt/β-catenin signaling pathway were significantly associated with a lower disease control rate and shorter median progression-free survival (PFS) and overall survival (OS) [22]. Abnormalities in the Wnt/β-catenin signaling pathway may influence the response of HCC to ICI therapy, as this pathway has been shown to create an “immunologically cold” TME by reducing T lymphocyte infiltration [23,24]. Genomic analyses indicate that the *Catenin Beta 1* (*CTNNB1*) gene, which encodes β-catenin, exhibits gain-of-function mutations, while the *axis inhibition protein1* (*AXIN1*) gene shows loss-of-function mutations in 15% to 35% of HCC patients [25,26,27]. In gene-engineered HCC mouse models, inducing exogenous *myelocytomatosis oncogene* (*MYC*) expression and *Trp53-/-*antigen upregulated the β-catenin pathway, thereby driving resistance to nivolumab and pembrolizumab through immune escape [28]. Recent findings also indicate that *JAK1/2* mutations in tumor cells inhibit the upregulation of PD-L1 expression, leading to primary resistance to PD-1 therapy [29]. The abnormal activation of the signal transducer and activator of transcription 3 (STAT3) signaling pathway in HCC also enhances the resistance to ICIs. The abnormal activation of STAT3 leads to an increase in the expression of cytokines such as transforming growth factor-β (TGF-β), Interleukin 17 (IL-17) and vascular endothelial growth factor (VEGF), while inhibiting the production of type I interferon (IFN-I) and the activity of natural killer (NK) cells, and promoting the immunosuppressive state of the TME [30].

#### 2.1.4. T Cells Dysfunction

CD8+ cytotoxic T cells are pivotal in the anti-tumor immune response. However, prolonged antigen exposure, as seen in chronic infections and cancers, can impair the function and development of these T cells, ultimately leading to functional exhaustion [31,32]. This exhaustion is characterized by the upregulated expression of multiple inhibitory receptors, such as PD-1, cytotoxic T lymphocyte-associated antigen-4 (CTLA-4), T cell immunoglobulin and mucin domain-containing protein 3 (TIM-3), Lymphocyte activation gene-3 (LAG-3), and herpes virus entry mediator (HVEM), along with reduced secretion of IFN-γ, tumor necrosis factor-α (TNF-α), granzyme A, and granzyme B [33].

The *CT10 regulator of kinase-like* gene (*CRKL*) is highly expressed in various cancer tissues and is closely associated with tumor cell proliferation, migration [34], and resistance to EGFR inhibitors [35,36]. Increased CRKL expression has also been observed in HCC patients who do not respond to anti-PD-1 therapy. Animal studies have shown that in tumor tissues with *CRKL* overexpression, the proportion of activated CD8+ T cells decreases, while the proportion of functionally exhausted CD8+ T cells significantly increases [37]. Mechanistically, CRKL inhibits adenomatous polyposis coli-mediated proteasomal degradation of β-catenin by competitively decreasing AXIN1 binding, thereby promoting vascular endothelial growth factor A (VEGF-A) and C-X-C motif chemokine ligand 1 (CXCL1) expression. This results in PD-L1-positive tumor-associated neutrophils (TANs) infiltration, which blocks the infiltration and function of CD8+ T cells [37].

In the resistant mouse model, HCC cells maintain the stability of poliovirus receptor (PVR) on the cell surface by upregulating the expression of poliovirus receptor-related 1 (PVRL1). Subsequently, PVR binds to the inhibitory receptor T cell immune receptor with Ig and ITIM domains (TIGIT) on the surface of T cells, transmitting inhibitory signals and ultimately leading to the exhaustion of CD8+ T cell function [38]. The expression level of TIGIT on CD8+ T cells in HBV-HCC patients is significantly elevated, and TIGIT+ PD-1+ CD8+ T cells are also upregulated. TIGIT+ PD-1+ CD8+ T cells exhibit characteristics of exhausted T cells, including high susceptibility to apoptosis and reduced cytokine secretion capacity [39]. The Tim-3/galectin-9 signaling pathway can also induce T cell dysfunction in individuals suffering from HBV-associated HCC [40,41].

Additionally, T cell dysfunction is associated with increased VEGF expression [42]. The combination therapy of ICIs and anti-VEGF treatment may improve therapeutic efficacy. This provides the preclinical rationale for the phase III IMbrave150 trial, which investigates the combined treatment of atezolizumab and bevacizumab versus sorafenib in patients with unresectable HCC who have not received prior systemic treatment. After a median follow-up of 8.6 months, the trial demonstrated statistically significant and clinically meaningful improvements in OS and PFS. Specifically, the median OS in the atezolizumab plus bevacizumab group was not reached, compared with 13.2 months in the sorafenib group. The median PFS was 6.8 months in the atezolizumab plus bevacizumab group versus 4.3 months in the sorafenib group [5]. This also provides a basis for the National Comprehensive Cancer Network (NCCN) guidelines to consider it as the standard treatment option for advanced-stage cases in the first-line of treatment.

#### 2.1.5. Immunosuppressive Cells in Microenvironment

A key feature of the liver microenvironment is the predominance of immunosuppressive cell types, including Kupffer cells, tumor-associated macrophages (TAMs), TANs, regulatory T cells (Tregs), and myeloid-derived suppressor cells (MDSCs) [43]. This immune-tolerant environment contributes to persistent liver damage resulting from chronic infection, leading to chronic hepatitis, liver cirrhosis, and, ultimately, HCC. Furthermore, this environment facilitates tumor cell immune escape, thereby promoting the initiation and progression of HCC [43,44]. The TME in HCC consists of a complex network of cancer cells, immune cells, cytokines, and extracellular matrix components. Most HCC patients exhibit limited immune infiltration and many immunosuppressive cells, contributing to immune resistance [45,46].

In HCC, macrophages, including resident Kupffer cells and those migrating from the blood circulation, can mediate anti-cancer responses such as phagocytosis and antigen presentation. However, they can also undergo functional reprogramming into TAMs [47,48]. The heterogeneous and plastic phenotype of macrophages can affect the balance between antitumor immunity and immune evasion in HCC [49,50]. M1-like TAMs play pro-inflammatory and anti-tumor roles, whereas M2-like TAMs exhibit anti-inflammatory and tumor-promoting functions. M2-type TAMs can facilitate carcinogenesis, suppress anti-tumor immune responses, stimulate angiogenesis, and enhance the invasion, migration, and intravasation of tumor cells [50,51]. In a mouse HCC model, a high proportion of M2-type TAMs exhibited stronger resistance to ICI treatment [52]. Zinc finger protein 64 (ZFP64) is upregulated in anti-PD-1-resistant HCC patients. At the mechanism level, the phosphorylation of ZFP64 can induce the transcriptional activation of colony-stimulating factor 1 (CSF1) in tumor cells, thereby promoting the recruitment and polarization of macrophages, enabling them to form the immunosuppressive M2 phenotype, and thereby mediating the resistance to anti-PD-1 therapy in HCC. The researchers further revealed that protein kinase Cα (PKCα) is the upstream kinase mediating the phosphorylation of ZFP64, and CSF1 is one of the key transcriptional downstream effectors of this signaling pathway in HCC cells. Inhibitors targeting the PKCα/ZFP64/CSF1 pathway can overcome anti-PD-1 resistance in HCC [53]. Similarly, the high expression of endogenous osteopontin (OPN) in HCC cells can also promote the polarization of macrophages to the M2-like phenotype, and upregulate the expression of PD-L1 by activating the CSF1/colony-stimulating factor receptor (CSF1R) signaling pathway in macrophages [54]. Furthermore, intratumoral palmitoyl-protein thioesterase 1 (PPT1) positive macrophages correlate with an immune-exhaustion contexture and exhibit an immunosuppressive phenotype in the HCC microenvironment. In tumor tissues with higher levels of PPT1+ macrophages, the level of CD4+ Tregs shows a mild upward trend, while the infiltration of CD8+ T cells significantly increases. However, the expression level of PD-1 in CD8+ T cells rises, suggesting that PPT1+ macrophages may play a potential role in promoting the functional exhaustion of CD8+ T cells. Additionally, enhanced checkpoint activity, T cell co-inhibition, APC co-inhibition, and an attenuated type II IFN-γ response were found in high-PPT1-expression HCC [55]. Anti-PD-1 therapy activates T cells to secrete IFN-γ, which induces IL-1β production in macrophages. IL-1β can upregulate Pim2 proto-oncogene serine/threonine protein kinase (PIM2) in HCC cells by activating the mitogen-activated protein kinase (MAPK) and nuclear factor kappa B (NF-κB) pathways. PIM2 upregulates anti-apoptotic proteins (Mcl-1 and Bcl-2), inhibiting T cell-mediated apoptosis of cancer cells [56]. Moreover, secreted phosphoprotein 1 (SPP1) positive macrophages interact with cancer-associated fibroblasts (CAFs) and tumor cells, promoting the formation of a tumor immune barrier (TIB) to prevent T cell infiltration [57]. CAFs are components of the tumor stroma, which can influence tumor progression and regulate the TME. PD-L1+ neutrophils induced by HCC-CAFs impair T cell function through the PD-1/PD-L1 pathway [58].

Beyond macrophages, other immunosuppressive cell populations also play pivotal roles in shaping the immune-tolerant TME of HCC and mediating therapeutic resistance, with Tregs being one of the most well-characterized subsets.

Tregs can inhibit the function of effector CD8+ T cells via inhibitory cytokines and direct cell contact [59,60]. A study has shown that in HCC, activated Tregs within the tumor exhibit high expression levels of various inhibitory molecules, including CTLA-4 and TNFRSF9. In patients with no response to anti-PD-1 combined with lenvatinib treatment, the proportion of activated Treg cells in the TME significantly increases. The enrichment of activated Treg cells is closely related to the TME and may be an important mechanism leading to the limited efficacy of this combined therapy [61].

In addition, neutrophils have emerged as key contributors to immunotherapy resistance in HCC, with specific subsets and their derived structures exerting distinct immunosuppressive effects.

A specific subset of neutrophils might drive resistance to anti-PD-1 therapy in patients with HCC [62,63]. The researchers further explored the relationship between neutrophils and anti-PD-1 resistance using single-cell RNA sequencing. They found that CD10 positive and alkaline phosphatase (ALPL) positive neutrophils were significantly enriched in patients with HCC who were resistant to anti-PD-1 treatment. These neutrophils can effectively inhibit the anti-tumor activity of CD8+ T cells in HCC patients and promote the irreversible functional exhaustion of T cells, thereby causing the tumor to develop resistance to anti-PD-1 treatment [63]. Neutrophil extracellular traps (NETs) are a network structure released by neutrophils and are an important component of the body’s innate immune system. NETs play a role in capturing and eliminating pathogens in the anti-infection immune response while also participating in protecting tumor cells and promoting thrombosis in the TME. Studies have shown that NETs can activate the Notch2-mediated NF-κB signaling pathway, upregulate the expression of CD73, promote the infiltration of Tregs, and thereby mediate the immune escape of HCC [64]. Targeting NETs is expected to become a potential strategy for enhancing the efficacy of immunotherapy.

Similar to macrophages, neutrophils, and Tregs, MDSCs are critical regulators of the immunosuppressive TME in HCC, with their expansion and functional activation closely linked to tumor progression and immunotherapy failure.

Clinical data indicate that peroxisome proliferator-activated receptor-gamma (PPARγ) is upregulated in the tumor in 40% of patients resistant to pembrolizumab, and high PPARγ expression is associated with a lower survival rate [57]. Mechanistically, resistant HCC cells activate VEGF-A via the upregulation of PPARγ transcription, driving the expansion of MDSCs [57]. This mechanism may provide a theoretical basis for the adjuvant therapy regimen of atezolizumab combined with bevacizumab as the current preferred treatment option. Additionally, MDSCs can promote the polarization of TAMs to the M2 phenotype [46], facilitate the formation of Tregs [65], and induce the differentiation of fibroblasts into CAFs [66]. Higher MDSC infiltration is associated with a poor clinical prognosis in HCC patients [67].

#### 2.1.6. Cytokines

TGF-β, IL-10, and VEGF within the TME all contribute to immunosuppression. TGF-β is abundantly present in the TME of HCC, primarily secreted by tumor cells, TAMs, and Tregs. It inhibits anti-tumor immune responses at multiple levels. Its mechanisms of action include inducing macrophages to polarize toward the pro-tumor M2 phenotype [68], promoting the differentiation of naive CD4+ T cells into Tregs [69], and weakening the cytotoxic functions of effector CD8+ T cells and NK cells [70,71]. Studies have shown that elevated levels of TGF-β in the serum of HCC patients may indicate a poorer response to pembrolizumab treatment [72]. IL-10, secreted by tumor cells, TAMs, Tregs, and DCs [73], can inhibit the recruitment and activity of tumor-infiltrating T cells [74,75], upregulate the expression of PD-L1 in monocytes [76], and promote the aggregation of MDSCs [77]. VEGF, primarily secreted by tumor cells and the surrounding stroma [78], promotes angiogenesis and weakens the anti-tumor response by exerting negative effects on APCs and effector T cells. Additionally, VEGF maintains immune tolerance in the TME by actively increasing the recruitment of MDSCs and Tregs [79]. This indicates that, in addition to the effectiveness of immunotherapy combined with anti-VEGF drugs in HCC, treatments targeting TGF-β and IL-10 may also be combined with immunotherapy in the future to enhance immunotherapeutic efficacy and provide more treatment options for patients.

#### 2.1.7. Exosomes

Exosomes, such as circular RNAs, can mediate the resistance of HCC to immunotherapy through mechanisms such as regulating the TIME and influencing the functions of immune cells. For example, HCC cells regulate the expression of CD39 in macrophages via circTMEM181 in exosomes. The CD39 on macrophages and the CD73 on HCC cells cooperatively activate the adenosine pathway, thereby inhibiting the function of CD8+ T cells, as evidenced by reduced secretion of granzyme B and increased expression of PD-1 and TIM-3 [80]. Circular RNA cell division cycle and apoptosis regulator 1 (circCCAR1), as exosomes, released by HCC cells are taken up by CD8+ T cells, which can inhibit the proliferation of activated CD8+ T cells, promote their apoptosis, reduce their cytotoxicity and cytokine secretion capacity, and up-regulate the expression of LAG-3, PD-1, TIM-3 and TIGIT on their surface, inducing CD8+ T cell dysfunction. CD8+T cell exhaustion mediated by exosome circCCAR1 can reduce the effective targets of the PD-1 pathway, thereby leading to primary resistance of HCC to anti-PD-1 therapy [81]. Ubiquitin-like PHD and RING finger domain 1 (UHRF1) is a major regulator of DNA methylation and is overexpressed in HCC and other cancers. Upregulated UHRF1 expression promotes the progression and tumorigenesis of HCC [82]. CircUHRF1 inhibits the secretion of TNF-α and IFN-γ from NK cells [13]. Additionally, it increases the expression level of TIM-3 by degrading miR-449c-5p, thereby inhibiting the function of NK cells and leading to resistance to anti-PD-1 treatment [13].

#### 2.1.8. Metabolic Reprogramming

Metabolic reprogramming describes the capacity of cells to alter their metabolic pathways in response to energy and biosynthetic requirements, supporting proliferation and enhancing resilience under stressful conditions [83]. Tumor cells exhibit vigorous proliferative characteristics, and their metabolic networks undergo significant reprogramming, which manifests as enhanced processes such as glucose uptake, nutrient utilization, and synthesis, storage, and conversion of adenosine triphosphate (ATP). Key metabolic pathways, such as glycolysis, fatty acid synthesis, and amino acid metabolism, are highly active, aiming to provide sufficient biosynthetic precursors and energy substrates for cancer cells to meet their rapid growth requirements and enhance their ability to adapt to environmental stress [84,85].

The Warburg effect represents a key mode of glucose metabolism in cancer cells, characterized by high glucose consumption and substantial lactic acid production even under oxygen-sufficient conditions [86]. This effect has significant and diverse impacts on various biosynthetic pathways, biological processes, and the processes responsible for generating signaling metabolites that promote cancer cell metastasis and immune evasion, thereby forming an immunosuppressive TME [87,88]. Researchers identified a novel protein encoded by circular RNA PETH (circPETH-147aa) and revealed its crucial role in the metabolic reprogramming of HCC. Norathyriol is a small-molecule inhibitor targeting circPETH-147aa. The results of in vivo experiments showed that the combination of anti-PD-1 antibody and norathyriol significantly improved the anti-tumor efficacy compared with the use of the anti-PD-1 antibody alone [89]. In patients with HCC, elevated serum levels of galectin-4 are associated with resistance to azacitidine/bevacizumab treatment and poor prognosis. The overexpression of galectin-4 leads to metabolic adaptation and the creation of an immunosuppressive TME, manifested by reduced CD8 T cell infiltration, impaired cytotoxicity, and increased accumulation of PD-L1 TANs. Mechanistically, galectin-4 inhibits proteasomal degradation of lactate dehydrogenase A (LDHA) by competitively reducing the 28 binding sites contained in the trimeric domain, thereby enhancing glycolysis and amplifying the expression of HIF-1α-mediated C-X-C motif chemokine ligand 6 (CXCL6). In preclinical models, reversing metabolic adaptation and immune exclusion via gene knockdown or drug inhibition of mannose protein-4 can restore sensitivity to azacitidine/bevacizumab treatment [90].

The reprogramming of fatty acid metabolism is associated with cancer immune evasion [91,92]. In immunotherapy-resistant liver cancer tissues, transforming acidic coiled-coil containing protein 3 (TACC3) is highly expressed and capable of reprogramming polyunsaturated fatty acid metabolism via acyl-CoA synthetase long-chain family member 4 (ACSL4), through pathways involving La-related protein 1 (LARP1) and poly(A)-binding protein cytoplasmic 1 (PABPC1). This metabolic shift limits the availability of crucial polyunsaturated fatty acids, such as docosahexaenoic acid (DHA), to CD8+ T cells, consequently impairing their ability to mount an effective anti-tumor response [93]. Riplet is an E3 ubiquitin ligase that specifically ubiquitinates the viral RNA sensor RIG-I, thereby promoting antiviral innate immune signaling and IFN-I production. Lack of Riplet leads to reprogramming of fatty acid metabolism in HCC cells, characterized by enhanced fatty acid synthesis mediated by fatty acid synthase (FASN), resulting in the accumulation of free fatty acids (FFAs) in the TME. The FFAs produced by liver cancer cells promote the terminal exhaustion of CD8 T cells through the STAT3 signaling pathway activated by CD36. Targeting FASN with specific inhibitors can reverse the exhausted T cell state and synergize with anti-PD-1 immunotherapy to exert anti-tumor effects [94].

### 2.2. Acquired Resistance

Acquired resistance refers to the scenario where ICI treatment initially shows efficacy, but the disease subsequently progresses or recurs clinically and/or radiologically [11]. This phenomenon can be attributed to two distinct underlying mechanisms shown in Figure 2A. First, it may stem from the presence of a heterogeneous tumor cell population and the subsequent selection of pre-existing resistant clones that were harbored in the tumor prior to the initiation of treatment. Second, it can result from the de novo acquisition of somatic mutations in tumors following immunotherapy exposure, which drives the development of unresponsiveness to immunotherapeutic agents [12].

In HCC, the phenomenon of acquired resistance is relatively common. A cohort study on gastrointestinal malignancies reported that in the hepatobiliary cancer subgroup, the incidence of acquired resistance to ICIs was 50% (14/28) [95]. A retrospective study found that among patients with unresectable hepatocellular carcinoma who received ICIs combined with lenvatinib treatment, 35.1% showed primary resistance and 36.6% developed acquired resistance. Patients with primary resistance tend to experience more rapid and extensive disease progression than those with acquired resistance [96].

Research on its mechanism is also fraught with difficulties due to the lack of unified diagnostic criteria, challenges in obtaining high-quality clinical samples, and the limited number of differentially expressed genes identified via extensive DNA/RNA sequencing [11]. Due to the insufficient research on the acquired ICI resistance mechanisms in HCC, we also reported the acquired resistance mechanisms that have been discovered in other malignant tumors.

#### 2.2.1. Decreased Antigen Presentation

Similar to primary resistance, dysfunction of the antigen-presenting system is also strongly associated with the development of acquired resistance in lung cancer and melanoma [97,98]. β2-microglobulin (B2M), a key molecule in the process of tumor antigen presentation, plays a crucial role in regulating the proper folding of MHC-I molecules and the loading of peptide segments by stabilizing the structure of MHC-I molecules on the cell surface [99]. In a study on the response of mismatch repair-deficient tumor patients to PD-1 blockade therapy, a total of five cases of drug resistance were observed, among which two patients were detected with *B2M* gene mutations [100]. However, there is almost no applicable evidence for HCC.

#### 2.2.2. Tumor Heterogeneity

The mechanism by which cancer cells lose the expression of neoantigens may lead to acquired resistance to ICIs. The interaction between tumor cells and the immune system during immunotherapy results in clonal selection within the tumor, favoring the retention of tumor cells with low levels of immunogenicity or neoantigen expression. This process ultimately triggers acquired resistance [101]. Anti-CTLA-4 antibodies can induce the upregulation of indoleamine 2,3-dioxygenase (IDO) expression in some HCC tumor cells, especially the IDO1 subtype. This process depends on the secretion of IFN-γ by T cells. Neutralizing IFN-γ can effectively block the upregulation of IDO expression. IDO catalyzes the degradation of tryptophan and promotes the production of immunosuppressive metabolites, inhibiting T cell function and thereby mediating tumor immune escape. In mouse HCC models, inhibiting IDO activity can restore the sensitivity of tumors that were previously resistant to anti-CTLA-4 treatment and significantly enhance the therapeutic effect [102]. A I/II phase clinical trial is evaluating the efficacy of the IDO1 inhibitor BMS-986205 combined with nivolumab as a first-line or second-line treatment option for patients with HCC (NCT03695250).

#### 2.2.3. Alternative Immune Checkpoints

Acquired resistance to ICIs is associated with the upregulation of alternative immune checkpoints, including TIM-3, CTLA-4, and lymphocyte activation gene 3 (LAG-3) [103,104,105]. The continuous exhaustion of T cells is positively correlated with the increased co-expression of PD-1, CTLA-4, TIM-3, LAG-3, and B And T lymphocyte associated (BTLA) [106], indicating that the co-expression of multiple immune checkpoints is associated with a severely exhausted T cell state. Δ42PD-1, a splicing variant of PD-1, has been observed to increase in proportion in patients with HCC who received treatment with nivolumab or pembrolizumab. These Δ42PD-1+ T cells highly express various exhaustion-related markers, such as VISTA, TIM-3, CD244, and LAG-3. Resistance to anti-PD-1 treatment occurs due to the inability of anti-PD-1 therapy to effectively target Δ42PD-1 [107]. A literature report described a case of a patient with primary HCC who developed acquired resistance to PD-1 inhibitors following surgical intervention. The treatment was subsequently switched to PD-L1 inhibitors, resulting in complete tumor remission [108]. The resistance may be related to the mutation of PD-1.

#### 2.2.4. Defects in IFN-γ Signaling

Effector T cells release IFN-γ, which triggers a signaling cascade in tumor cells via the JAK-STAT pathway, mediating the expression of MHC-I and PD-L1 and inducing tumor cell death through multiple mechanisms [109]. The crucial first step in this pathway is the activation of receptor-associated kinases JAK1 and JAK2 through the binding of IFN-γ to the heterodimeric IFNGR1/IFNGR2 [110]. Among four melanoma patients with acquired resistance to PD-1 blockade therapy, two were found to have loss-of-function mutations in the genes encoding IFN receptor-associated JAK1 or JAK2, accompanied by the deletion of the wild-type allele [97]. Excessive accumulation and prolonged activation of IFN-γ may lead to reduced expression of IFN-γ receptors and diminished responsiveness in HCC cells, facilitating tumor cell immune evasion [111]. Additionally, IFN-γ stimulates the secretion of IL-10 and VEGF and enhances PD-L1 expression, contributing to the emergence of acquired resistance to ICIs [112].

#### 2.2.5. Immune-Excluded Tumor Microenvironment

Researchers reported a case of a patient with HCC who received 15 cycles of atezolizumab combined with bevacizumab therapy. The best tumor response achieved was partial response. Subsequently, disease progression occurred, and the patient underwent surgical resection following progression.

Compared with baseline HCC tissues, two notable changes were observed in progressive tumor tissues. First, the expression level of PD-L1 in tumor-infiltrating immune cells and the number of tumor-infiltrating CD8+ T cells were both reduced. This finding suggested the formation of an immune-rejective TME. Second, the expression of hepatoblast characteristic-related genes in tumor cells was upregulated after progression. This indicated that tumor cells might have undergone dedifferentiation.

Collectively, the establishment of an immune-rejective TME and the dedifferentiation of tumor cells synergistically promoted the acquisition of resistance to atezolizumab plus bevacizumab therapy in this HCC patient [113].

**Table 1 cancers-18-00039-t001:** Summary of mechanisms of ICI resistance.

Resistance Mechanism	Type of Resistance	Data Source	Clinical Development Status	Ref.
EGFR Signaling Upregulation	Primary	HCC-Preclinical	Ongoing clinical trials of EGFR inhibitors	[21]
CRKL expression	Primary	HCC-Preclinical	Absence	[37]
Increased VEGF expression	Primary	HCC-Preclinical	Drugs available in HCC therapy	[114]
PVRL1 Upregulation	Primary	HCC-Preclinical	Absence	[38]
Increased VEGF expression	Primary	HCC-Strong clinical evidence (basis for atezo + bev)	Drugs Available: VEGF inhibitors already combined with ICI	[79]
Wnt/β-catenin Pathway Activation	Primary	HCC-Clinical (HCC genomic data); Preclinical (HCC)	Ongoing clinical trials in advanced solid tumors (NCT05919264)	[25,26,27]
JAK-STAT3 pathway activation	Primary	HCC-Preclinical	Ongoing clinical trials in ICI resistance NSCLC (NCT06925048)	[79]
Immunosuppressive Cell Infiltration (MDSCs, M2-TAMs, Tregs)	Primary	HCC-Strong clinical and preclinical evidence (high infiltration correlates with poor prognosis/resistance)	Ongoing trials: Targeting TAMs (e.g., CSF-1R inhibitors), MDSCs, or CAFs in combination with ICI is an active area of early-phase trials.	[51,53,59]
Exosomes(circRNAs)	Primary	HCC-Preclinical	Absence	[81]
Metabolic reprogramming (Warburg effect, fatty acid metabolism)	Primary	HCC-Preclinical	Absence	[89,90,94]
*B2M* gene mutation	Acquired	Lung cancer and melanoma-Clinical	Absence	[100]
IFN-γ Signaling defect	Acquired	Melanoma-Clinical	Absence	[97]
Alternative Immune Checkpoints (e.g., TIM-3, LAG-3, PD-1 isoform)	Acquired	HCC-Clinical (HCC tissue post-immunotherapy)	Ongoing Clinical trials: Next-generation ICI combinations (e.g., anti-PD-1 + anti-TIM-3/LAG-3 mAbs) are in early-phase HCC trials.	[107]
Immune-rejective tumor microenvironment	Primary/Acquired	HCC-Clinical (case report)	Absence	[113]
Tumor cell dedifferentiation	Acquired	HCC-Clinical (case report)	Absence	[113]

## 3. Biochemical Predictors of Response to ICIs in HCC

The main biomarkers for predicting the efficacy of traditional immunotherapy mainly refer to the expression level of PD-L1 in the TME, including the expression of PD-L1 in tumor cells and immune cells, as well as TMB and microsatellite instability (MSI). These three indicators are widely used as predictive biomarkers for the efficacy of immunotherapy in various tumors such as lung cancer, colorectal cancer, and melanoma. They have also been extensively explored in predicting the efficacy of liver cancer immunotherapy.

### 3.1. PD-L1

Numerous clinical studies have explored whether PD-L1 can serve as a biomarker for predicting the efficacy of immunotherapy in HCC. The Keynote-224 study reported the clinical efficacy of pembrolizumab for second-line treatment of advanced HCC and further analyzed the correlation between the baseline PD-L1 expression level in tumor tissues and the treatment response. The results showed that the PD-L1 expression level as reflected by the combined positive score (CPS), which is a comprehensive index incorporating the number of PD-L1-positive tumor cells and tumor-infiltrating immune cells, was significantly correlated with the treatment response to pembrolizumab; however, when only the PD-L1 expression on tumor cells was used as an indicator (tumor proportion score, TPS), no significant association with the clinical efficacy was observed [115]. The CheckMate 040 study confirmed this view, namely that there is no correlation between the baseline tumor PD-L1 expression level and the treatment response [116]. However, analysis in the CheckMate 459 and GO30140 studies revealed that PD-L1 expression is significantly correlated with treatment efficacy and patient prognosis. The higher the level of PD-L1 expression within the tumor, the higher the immune therapy response rate, and the better the long-term prognosis of patients [117,118].

It should be noted that PD-L1 expression in clinical trials has been quantified using different immunohistochemical scoring systems. TPS is defined as the percentage of viable tumor cells showing membranous PD-L1 staining among all viable tumor cells, whereas CPS is calculated as the number of PD-L1–positive tumor cells, lymphocytes and macrophages divided by the total number of viable tumor cells, multiplied by 100; CPS is therefore a composite metric that integrates both tumor and immune-cell staining [119]. By contrast, the immune cell (IC) score, used in several atezolizumab-based HCC studies, quantifies PD-L1 expression as the proportion of the tumor area occupied by PD-L1–positive tumor-infiltrating immune cells, and PD-L1 “positivity” has been explored at different IC or combined tumor-cell/immune-cell (TC/IC) cutoffs (e.g., ≥1%, ≥5% or ≥10%) [115]. In addition, different trials have adopted different thresholds to define PD-L1 positivity (e.g., TPS ≥ 1% vs. ≥25%, CPS ≥ 1 vs. ≥10, or IC ≥ 1–10%), even when evaluating the same agent, and have employed non-identical PD-L1 antibody clones (such as 22C3, 28-8, SP142, SP263 and E1L3N), which are known to differ in analytical sensitivity and staining patterns [120]. These methodological inconsistencies—including the use of TPS, CPS or IC scores, heterogeneous positivity cutoffs, and non-harmonized antibody clones and assay platforms—substantially complicate cross-trial comparisons and may, at least in part, account for the apparently contradictory observations regarding the predictive value of PD-L1 in HCC [121].

There is thus no clear consensus on whether PD-L1 can be used as a robust biomarker for predicting the efficacy of immunotherapy for HCC, and further standardized and prospectively designed studies are still needed.

### 3.2. Tumor Mutational Burden and Microsatellite Instability

A study conducted a systematic analysis of the genomic characteristics of 755 HCC samples and explored the correlations between several potential biomarkers and the response to PD-1 inhibitor therapy. The results showed that the median TMB of HCC samples was 4 mutations/Mb, and 95% of the samples had a TMB lower than 10 mutations/Mb, with only 1–2% of the samples showing a high tumor mutation burden (TMB-H). TMB-H was found in a small number of liver cancer patients, thus limiting the clinical applicability of TMB as a predictive biomarker for liver cancer [122]. Based on the results of the KEYNOTE-016 study, the US Food and Drug Administration (FDA) has approved pembrolizumab for the treatment of unresectable solid tumors in patients with MSI high (MSI-H) or mismatch repair deficiency (dMMR) status. However, since less than 3% of patients with HCC have the MSI-H status, the clinical application value of assessing MSI status in HCC may be relatively limited [122,123,124]. Further research is needed to evaluate its clinical value [100]. Due to the extremely low incidence of TMB-H and MSI-H in liver cancer patients, their use in predicting the efficacy of immunotherapy for liver cancer patients is limited and cannot guide the selection of clinical treatment plans [124]. Although the incidences of TMB-H and MSI-H are low in HCC, other genomic features, such as *CTNNB1* mutations, angiogenesis signatures, and inflammatory gene expression profiles, may hold greater predictive potential and are under active investigation.

### 3.3. Gut Microbiota

Multiple studies have shown that the gut microbiome may play a crucial role in mediating the efficacy of ICIs [125,126,127]. The gut microbiota is believed to promote the malignant transformation of liver cirrhosis into HCC and has significant bidirectional interactions with the host immune system, capable of regulating the body’s immune response [128,129]. A study found that a lower proportion of the *Prevotella phylum*/*Bacteroides* phylum can serve as a biomarker for the lack of response of HCC patients to nivolumab, while the presence of the *Akkermansia genus* is associated with a favorable treatment response [130]. The dynamic variation characteristics of the gut microbiome can provide early prediction for the outcome of HCC immunotherapy [131]. New insights into the pathological and physiological correlations between gut microbiota dysbiosis and the prognosis of HCC may lead to innovative treatment solutions, such as supplementing probiotics to prevent primary resistance to treatment.

### 3.4. Other Emerging Biomarkers

In addition to the existing multiple predictive indicators based on histology, genomics and microbiomics, a series of emerging biomarkers with more diverse sources and potential predictive value have gradually been proposed in clinical research. The researchers found that after 6 weeks of anti-PD-1 immunotherapy, a reduction of more than 50% in AFP or abnormal prothrombin (PIVKA-II) levels compared to the baseline was significantly associated with a better therapeutic response and a longer survival period [132]. A study found that a higher baseline plasma TGF-β level (≥200 pg/mL) was associated with no response to pembrolizumab treatment [72]. Clinical studies have shown that inflammatory-related biomarkers are closely related to the non-response and prognosis of immunotherapy for HCC [133], including indicators such as the neutrophil-to-lymphocyte ratio (NLR) [134,135], platelet-to-lymphocyte ratio (PLR) [134], and C-reactive protein (CRP) [136]. Moreover, HCC patients with a lower albumin-bilirubin (ALBI) grade show better treatment responses and better prognosis to immunotherapy [137].

## 4. Combined Therapy to Overcome ICI Resistance

### 4.1. Combined with Molecular Targeted Therapy

There is no unified method for managing drug resistance. Existing guidelines recommend the use of TKIs following first-line treatment with atezolizumab combined with bevacizumab [138]. The Imbrave251 phase III study is currently evaluating atezolizumab combined with lenvatinib or sorafenib compared with lenvatinib or sorafenib alone in treating HCC after progression on atezolizumab combined with bevacizumab (NCT04770896). Compared with ICIs monotherapy, patients receiving ICIs combined with bevacizumab or lenvatinib have a significantly lower rate of primary resistance [139]. Combining anti-PD-1 therapy with lenvatinib increases the response rate in HCC patients from 17% with anti-PD-1 monotherapy to 45% [115,140,141].

VEGF-targeted therapy inhibits the aberrant proliferation of the tumor vasculature and enhances CD8+ T cell infiltration within the TME, while suppressing the activity of immunosuppressive cells, such as TAMs, MDSCs, and Tregs. This mechanism contributes to the reversal of the immunosuppressive TME in HCC and helps overcome the development of resistance to ICIs [142,143]. In patients with advanced HCC, the combination of atezolizumab (an anti-PD-L1 monoclonal antibody) and bevacizumab (an anti-VEGF-A monoclonal antibody) has demonstrated better efficacy than sorafenib, establishing a new first-line treatment standard with a median OS of 19 months, marking a significant advancement in the treatment of HCC [144]. A study has found that in the ICIs combination treatment regimen, patients receiving ICIs combined with bevacizumab show a lower rate of acquired resistance. This is consistent with the persistent response trend observed in the IMbrave150 study, where atezolizumab was combined with bevacizumab [139]. A phase III trial (NCT04985136) on the efficacy of camrelizumab combined with apatinib in treating HCC patients who have failed PD-1 inhibitor therapy is currently underway (more ongoing trials in Table 2).

### 4.2. Combined with Local Treatment

RFA, microwave ablation (MWA), TACE, radiotherapy, and yttrium-90 (Y-90) radioembolization are all local therapies aimed at shrinking and controlling tumors within the liver. These approaches make the remaining cancer cells more susceptible to the immune system, which can then be enhanced with immunotherapy [145,146]. Liver-directed treatments that destroy tumor cells release tumor antigens, which can work in synergy with ICIs to achieve a better anti-tumor response. Compared with monotherapy using ICIs, patients receiving a combination of ICIs and local treatments have a lower risk of primary and secondary resistance [139].

A study has shown that combining RFA with anti-PD-1 monotherapy can increase the ORR of patients from 10% to 24%, suggesting that the combination of ablation therapy and immunotherapy has feasibility and potential clinical value for patients who show stable disease or atypical progression during anti-PD-1 monotherapy [147]. The results of another study showed that compared with RFA treatment alone, the combined treatment could extend the median recurrence-free survival period (RFS) of patients with recurrent HCC by 7.2 months, representing an increase of 87.8% [148].

After treatment with TACE, the residual tumor tissues of HCC show high PD-L1 expression [149,150]. TACE can also induce tumor cell necrosis, promote the release of tumor antigens, enhance the tumor-specific CD8+ T cell immune response, and inhibit the proliferation of Tregs, thereby converting “cold tumors” to “hot tumors”, to a certain extent [151,152]. The above mechanisms suggest that TACE may have the potential to enhance the efficacy of ICIs. A retrospective analysis demonstrated that combining nivolumab with TACE, as opposed to nivolumab monotherapy, exhibits a favorable safety profile and significantly prolongs PFS (8.8 months versus 3.7 months) [153]. A study evaluated the efficacy of PD-1 inhibitors combined with TACE in the treatment of unresectable HCC. The results showed that the ORR reached 42.9%, the disease control rate (DCR) was 90.5%, the 6-month PFS was 62.2%, and the median PFS was 7.5 months (95% CI: 5.76–9.23) [154]. Compared with the ORR of 15–20% and 18.4% for nivolumab and pembrolizumab monotherapy in advanced HCC in the CheckMate 040 [116] and KEYNOTE-240 [8] studies, the ORR of the combined treatment was significantly improved.

Radiation can promote anti-tumor responses by enhancing tumor antigen presentation, such as upregulation of MHC-I expression in tumor cells after external beam irradiation [155]. The combination of PD-L1 or PD-1 inhibitors with radiotherapy has been shown to elicit a stronger anti-tumor response in preclinical models [156]. A real-world study has shown that the median PFS for the group receiving combined radiotherapy and PD-1 inhibitors was 5.7 months, and the median OS was 20.9 months, significantly better than the group treated with PD-1 inhibitors alone (with a median PFS of 2.9 months and a median OS of 11.2 months) [146]. Y-90 radioembolization can also enhance the anti-tumor effect of ICIs by activating tumor-specific cytotoxic T cells via increased inflammatory co-signaling and tumor antigen presentation [157].

For patients with advanced HCC, the treatment of Y90 radioembolization combined with nivolumab showed an ORR of 30.6% [158]. This result is superior to the approximately 20% ORR when using Y90 radioembolization alone, and the reported values of 15% to 20% when using anti-PD-1 drugs alone [8,116].

### 4.3. PD-1/PD-L1 Inhibitors Combined with CTLA-4 Inhibitors

Ipilimumab is a monoclonal antibody targeting CTLA-4, which blocks the binding of CD80/CD86 ligands on APCs to the CTLA-4 receptor on activated T cells. This action eliminates the immune inhibitory signal, promoting the activation and clonal expansion of T cells [159]. When used in combination with nivolumab, a PD-1 inhibitor, it synergistically enhances the anti-tumor function of effector T cells. Additionally, ipilimumab can induce the lysis of Tregs via antibody-dependent cell-mediated cytotoxicity (ADCC), reducing their infiltration in the TME and further enhancing the anti-tumor activity of the combination therapy [159]. In the CheckMate 040 trial, the combination of nivolumab and ipilimumab was evaluated in patients who had received sorafenib treatment [160]. Tremelimumab, by blocking CTLA-4, promotes the activation and proliferation of T cells. Durvalumab, as a monoclonal antibody targeting PD-L1, specifically binds to PD-L1 on the surface of tumor cells, blocking its interaction with PD-1, thereby restoring the recognition and killing function of T cells against tumor cells. The combined use of these two can synergistically enhance the anti-tumor immune response and effectively increase the response rate of immunotherapy. In a clinical trial of tremelimumab combined with durvalumab as a second-line treatment for advanced HCC, an ORR of 22.7% was observed [161]. A phase III clinical trial compared the efficacy and safety of two drug combination regimens with sorafenib as the first-line treatment for advanced HCC [162]. The results demonstrated that the risk of death in the combination therapy group was reduced by 22% compared with sorafenib, with an ORR of 20.1%, and a median OS of 16.5 months. Based on these data, the FDA approved the combination of durvalumab and tremelimumab for the first-line treatment of unresectable HCC [114]. A randomized, open-label, controlled multicenter phase III clinical study (NCT04720716) is currently ongoing to test the efficacy of IBI310, an anti-CTLA-4 monoclonal antibody, in combination with the PD-1 inhibitor sintilimab as a first-line treatment for advanced HCC (details in Table 2).

**Table 2 cancers-18-00039-t002:** Combined treatment with immune checkpoint inhibitors in HCC.

Category	Agents	Target	Combinational Therapies	Therapy Line/Setting	Trial Number	Status	Ref.
Combined with molecular targeted therapy	Nimotuzumab	Anti-EGFR antibody	Anti-PD-1 or anti-PD-L1	Second-line	Phase II **NCT06413017**	Recruiting	NA
	Rivoceranib/apatinib	VEGFR2-targeted TKI	Camrelizumab	Post-ICIs	Phase III **NCT04985136**	Terminated	NA
	Rivoceranib/apatinib	VEGFR2-targeted TKI	Camrelizumab	First-line	Phase III NCT03764293	Completed	[163]
	Lenvatinib	TKI	Atezolizumab	Second-line (post- atezo + bev)	Phase III **NCT04770896**	Active, not recruiting	NA
	Bevacizumab	VEGF-targeting	Atezolizumab	First-line	Phase II **NCT04829383**	Active, not recruiting	NA
	Rivoceranib/apatinib	VEGFR2-targeted TKI	Camrelizumab	Post-ICIs	Phase II **NCT04826406**	Recruiting	NA
Combined with local treatment	TACE	NA	ICIs	ICI before/within 2 months after TACE	NCT04975932	Completed	[164]
	TACE	NA	PD-1/PD-L1 inhibitors + VEGF-TKI	First-line	A target trial emulation study NCT05332821	Unknown status	[165]
	HAIC	NA	PD-1 inhibitors + lenvatinib	Fiest-line	**NCT06632106**	Active, not recruiting	NA
	Incomplete thermal ablation	NA	Anti-PD-1	Second-line (post- sorafenib)	NCT03939975	Completed	[147]
	Radiation therapy	NA	First-line PD-1 therapy	Second-line (post-anti-PD-1)	Phase II **NCT06870942**	Recruiting	NA
	Particle beam radiation therapy	NA	ICIs	First-line	Phase II **NCT06828380**	Recruiting	NA
	Photon radiotherapy	NA	Tremelimumab + Durvalumab	First-line	Phase II **NCT06999707**	Recruiting	NA
	Radiofrequency ablation	NA	* Carrizumab	First-line	**NCT04150744**	Recruiting	NA
	Y90 radioembolization	NA	Nivolumab	Post-ICIs	Phase II NCT03033446	Active, not recruiting	[158]
	Y-90 SIRT	NA	Durvalumab and tremelimumab	First-line	Phase II **NCT04522544**	Recruiting	NA
	Y-90 TARE	NA	Atezolizumab and bevacizumab	First-line	Phase II **NCT04541173**	Terminated	NA
Multiple ICIs	Ipilimumab	CTLA-4 antibody	Nivolumab	First-line	Phase III NCT04039607	Active, not recruiting	[166]
	Ipilimumab	CTLA-4 antibody	Nivolumab	Second-line (post- atezo + bev)	Phase II **NCT05199285**	Terminated	NA
	Tremelimumab	CTLA-4 antibody	Durvalumab	First-line	Phase I/II NCT02519348	Active, not recruiting	[161]
	Tremelimumab	CTLA-4 antibody	Durvalumab	First-line	Phase III NCT03298451	Active, not recruiting	[167,168]
	IBI310	CTLA-4 antibody	Sintilimab	First-line	Phase III **NCT04720716**	Unknown status	NA

* Carrizumab is a novel PD-L1 inhibitor candidate. The bolded clinical trial numbers represent ongoing clinical trials, not completed, or whose results have not been officially published (the Ref. column is marked as NA). Abbreviations: ICIs, immune checkpoint inhibitors; EGFR, epidermal growth factor receptor; HAIC, hepatic arterial infusion chemotherapy; VEGFR, vascular endothelial growth factor receptor; HCC, hepatocellular carcinoma; TKI, tyrosine kinase inhibitor; TACE, transcatheter arterial chemoembolization; 90 Y-SIRT, yttrium-90 selective internal radiation therapy; Y-90 TARE, transarterial radioembolization with yttrium-90; CTLA-4, cytotoxic T-lymphocyte antigen 4; post- atezo + bev: post–atezolizumab plus bevacizumab treatment.

## 5. Potential Therapeutic Strategies

### 5.1. Enhancing T Cell-Mediated Immunity

One of the most promising approaches to improving T-cell responses against tumors is through tumor-specific vaccination, which helps to activate and enhance T-cell recognition. The vaccine works by enhancing the tumor-specific T cell response through active immunity. It mainly targets alpha-fetoprotein (AFP) and glypican 3 (GPC3) in HCC [169]. AFP is an important tumor-associated antigen for HCC and a potential immunotherapy target. Studies have shown that AFP immunization effectively induces AFP-specific CD8+ T cells in the body. However, the CD8+ T cells exhibit exhaustion characteristics. The AFP vaccine can prevent the occurrence of HCC, but it is ineffective against tumors that have already formed [170]. The cMet/β-catenin mouse model of HCC exhibits immune escape characteristics and shows resistance to traditional immunotherapy. In this model, neither anti-PD-1 nor anti-PD-L1 monotherapy has shown significant efficacy; however, AFP immunotherapy combined with anti-PD-L1 can significantly inhibit the progression of HCC, while the combination with anti-PD-1 can induce a slowdown in tumor progression [170]. Neoantigen vaccines can also be used to activate tumor-specific T cells. Relevant studies and details can be found in Table 3.

### 5.2. Targeting Signaling Pathway

In addition to enhancing T-cell recognition, targeting key signaling pathways that mediate tumor immune resistance is a critical approach. The CRKL/β-catenin/VEGF-A and CXCL1 signaling axes play a crucial role in the immune-resistance of HCC. Blocking CRKL/β-catenin/VEGF-A and CXCL1 axis using bevacizumab or lenvatinib effectively overcomes anti-PD-1 resistance. Activation of the CRKL/β-catenin/VEGF-A and CXCL1 axis is a critical obstacle to successful anti-PD-1 therapy. Therefore, CRKL inhibitors combined with anti-PD-1 could be useful for the treatment of HCC [37]. Furthermore, β-catenin is often mutated in HCC and plays a central role in establishing an immune-excluded TME. Targeting β-catenin signaling with siRNA delivered via nanoparticles has been shown to enhance anti-PD-1 treatment efficacy, making it a promising area for future combination therapies [176]. BC2059 is a β-catenin inhibitor and is currently in the early stage of clinical trials (details in Table 3).

### 5.3. Targeting Alternative Immune Checkpoints

In mouse models, the TIGIT pathway inhibits the cytotoxic function of CTLs, thereby enhancing the resistance of HCC cells to PD-1 inhibitors. However, the combined treatment strategy of PD-1 inhibitors and TIGIT inhibitors can increase the number of tumor-infiltrating CTLs, significantly inhibit tumor growth, and prolong the survival of mice [38]. Currently, preclinical and clinical studies are investigating the blockade of other immune suppressive checkpoints, such as LAG-3, TIM-3, and TIGIT (clinicaltrials.gov, accessed on 1 November 2025).

### 5.4. Targeting Suppressive Tumor Microenvironment

Reconstructing the tumor immune microenvironment is a strategy to enhance the efficacy of ICIs (Table 3 and Figure 3).

Additionally, glycogen synthase kinase 3 alpha (Gsk3a) inhibited the function of CTLs by inducing neutrophil chemotaxis. The combination treatment of the Gsk3a inhibitor (SB216763) and the anti-PD-1 antibody significantly inhibited tumor growth and enhanced immune response in a mouse orthotopic tumor model [177].

IL-8 has been demonstrated to recruit MDSCs and enhance immune escape in cancer via C-X-C motif chemokine receptor 2 (CXCR2) signaling [178]. Inhibiting the IL-8/CXCR2 pathway can significantly eliminate MDSC trafficking and immunosuppressive activity, thereby overcoming resistance to ICIs [179]. The selective PPARγ antagonist T0070907, when combined with anti-PD-L1 therapy, significantly reduced drug-resistant tumors and prolonged survival, while improving the immune status of the TME by increasing the proportion of cytotoxic T cells and decreasing the proportion of MDSCs [57].

Pharmacological suppression of PPT1 by DC661 led to reduced activation of the MAPK pathway while simultaneously inducing the NF-κB pathway in macrophages. Treatment with DC661 significantly improved the effectiveness of anti-PD-1 antibody therapy in a mouse model of HCC [55]. A phase II clinical trial of the use of GT90001 to block TGF-β receptors, in combination with nivolumab for patients resistant to ICIs, is also set to commence shortly. Carbonic anhydrases XII (CAXII) is highly expressed in tumor-infiltrating monocytes/macrophages and can affect immune cell functions by regulating the CCL8 and PERK signaling pathways [180]. CAXII inhibitors can regulate macrophage polarization, reduce the proportion of M2-type macrophages, increase the number of M1-type macrophages, and promote the infiltration of CD8+ T cells into tumors. In summary, CAXII inhibitors can reverse the immunosuppressive TME, thereby enhancing the therapeutic effect of PD-1/PD-L1 inhibitors. The results of in vivo experiments show that compared with using CAXII inhibitors alone or anti-PD-1 antibodies alone, the combination significantly enhances the anti-tumor effect of anti-PD-1 on HCC [87]. In addition to regulating the polarization direction of TAMs, the immunosuppressive microenvironment can be improved by directly targeting M2-type macrophages. Multiple studies have confirmed that M2-type macrophages play a crucial role in tumor immune evasion and resistance to ICI therapy [53,181]. Targeting key regulators involved in M2 phenotypic polarization, a process linked to immunosuppression and tumor progression, may enable therapeutic reprogramming of TAMs, thereby alleviating immunosuppressive conditions and improving responsiveness to ICI therapy. The CSF1/CSF1R signaling pathway plays a crucial role in mediating the polarization of macrophages to the M2 type [53]. Therefore, targeting this pathway is expected to enhance the therapeutic effect of ICIs.

PLX3397 is designed to target CSF1R. In the orthotopic liver cancer mouse model constructed by overexpressing OPN in HCC cells, the PLX3397 combination therapy significantly enhanced the therapeutic effect of anti-PD-L1 compared with the control group and the group treated with anti-PD-L1 alone, manifested by a reduced incidence of lung metastasis and prolonged survival period [54]. An ongoing clinical trial is evaluating the efficacy of PLX3397 combined with pembrolizumab in treating primary or secondary PD-1/PD-L1 resistant melanoma (NCT02452424).

### 5.5. Targeting Metabolic Reprogramming

Targeting specific metabolic checkpoints, such as glucose metabolism, fatty acid oxidation, or amino acid metabolism (glutamine, arginine, tryptophan), has shown promise in restoring immune cell function and enhancing immune responses [182]. Based on the above, FASN is associated with tumor immune escape. In preclinical HCC models, pharmacological inhibition of FASN restores T-cell effector function, increases MHC I expression, enhances tumor antigen presentation, and cooperates with PD-1/PD-L1 blockade (details see Table 3) [173]. In colon adenocarcinoma and melanoma, the combined application of anti-PD-1 antibody immunotherapy and lactate transporter inhibitors successfully delayed tumor growth. Compared with the use of anti-PD-1 antibody therapy alone, the combined therapy significantly increased the number of CD8 T cells expressing IFNγ within the tumors.

## 6. Discussion

Immunotherapy has rapidly advanced across various cancers, significantly improving patient survival and enabling some patients to achieve long-term tumor control. However, some patients in clinical practice do not respond to ICI treatment, or although they show a therapeutic response in the initial stage, they eventually develop acquired resistance. The first-line systemic treatment for HCC is recommended based on several key clinical studies, including atezolizumab combined with bevacizumab in the IMbrave150 study [144], sintilimab combined with bevacizumab biosimilar (IBI305) in the ORIENT-32 study [183], tremelimumab plus durvalumab in the HIMALAYA study, and camrelizumab combined with rivoceranib in the CARES-310 study [163]. In Imbrave150, atezolizumab plus bevacizumab achieved an ORR of 27.3%, a median progression-free survival (PFS) of 6.8 months, and a median OS of 19.2 months [5,144]. In ORIENT-32, sintilimab plus IBI305 demonstrated an ORR of 25.0%, a median PFS of 4.6 months, and a significant OS benefit compared with sorafenib [183]. In the HIMALAYA trial, the STRIDE regimen (tremelimumab plus durvalumab) yielded an ORR of 20.1% and a median OS of 16.4 months [114]. Similarly, in CARES-310, camrelizumab plus rivoceranib achieved an ORR of 25.4%, a median PFS of 5.6 months, and a median OS of 22.1 months [163]. However, the ORR of these studies is mostly between 20% and 40%, and the PFS is only 4 to 8 months, suggesting that the efficacy of some patients is difficult to maintain for a long time, usually developing resistance after 4 to 8 months of treatment. The OS of such patients is also not satisfactory, with a median OS of approximately 20 months. Therefore, exploring the mechanisms of ICI resistance and the corresponding solutions can benefit more patients. At present, there is a certain understanding of the primary resistance mechanism of HCC. However, the research on secondary resistance is still relatively scarce. The reasons for this include the lack of a precise definition of acquired resistance and the difficulty in obtaining tumor samples for analysis [11].

There are several difficulties in researching ICI resistance in HCC. First, the complexity of the TME remains a significant obstacle, as the interactions between different cell types within the TME are not yet fully elucidated, complicating the development of effective treatment strategies [184,185,186]. Second, the high genetic and epigenetic heterogeneity of HCC leads to considerable variability in drug responses among patients, making it challenging for existing treatments to produce consistent effects across all HCC subtypes [187]. Third, although the search for biomarkers is intensifying, no widely recognized biomarker reliably predicts the efficacy of immunotherapy. Future research on predictive biomarkers should focus more on the intrinsic factors of tumors and explore them in combination with new detection technologies. The intrinsic factors of tumors can be obtained using high-throughput technologies such as genomics, transcriptomics, or proteomics, mainly including mutant genes, copy number variation, gene insertions or deletions, gene signatures, tumor antigenomes, TCR clonality, and immune infiltrating cells in the TME.

Additionally, for patients whose T cell function is inactivated due to the suppression of the immune microenvironment, adoptive T cell therapy can be considered, such as CAR-T cell therapies. However, the application of CAR-T cells in HCC faces hurdles such as the tumor’s immunosuppressive microenvironment, which includes Tregs, MDSCs, and TAMs, all contributing to immune evasion. Despite these challenges, CAR-T cell therapy holds potential for overcoming some of these immune barriers, particularly in patients with suppressed T cell function due to the TME [188]. Research suggests that CAR-T cells, targeting specific tumor-associated antigens such as AFP and GPC3, can be effective in enhancing T cell responses [189]. Furthermore, combining CAR-T therapy with other treatments like ICIs or localized therapies such as RFA and TACE may improve the therapeutic outcomes.

Oncolytic virus therapy and mRNA-based personalized vaccines represent two rapidly evolving immunotherapeutic strategies with strong potential to enhance T-cell recognition of tumors, particularly in immunologically “cold” cancers [190,191]. Emerging clinical data indicate that mRNA vaccines can induce robust and durable CD8^+^ T-cell responses and may enhance the depth and durability of response to checkpoint blockade [192]. Both oncolytic virus therapy and mRNA-based personalized vaccines address fundamental mechanisms of immune resistance by enhancing tumor antigen availability and improving T-cell priming. As such, they represent promising components of future combination strategies aimed at overcoming immune escape and improving long-term outcomes in HCC and other solid tumors.

## 7. Conclusions

There are many studies on combined treatments with ICIs. Among them, atezolizumab combined with bevacizumab and tremelimumab combined with durvalumab have become the first-line treatment options for HCC in systemic therapy. However, the effect is still limited. Exploring new combined treatment regimens and other intervention strategies is significant for reversing ICI resistance. In future research, new strategies for reversing immune resistance include enhancing the immune system’s ability to recognize tumor antigens, as well as targeting and regulating the inhibitory state of the immune microenvironment. Promising strategies for enhancing tumor antigen recognition include oncolytic virus therapy and mRNA individualized vaccines. Targeted regulation of the suppressive state of the immune microenvironment can be achieved by acting on immune-suppressive cells, cytokines, their receptors, etc. Further exploration of the mechanisms underlying ICI resistance in HCC will help to develop more effective treatment strategies, enhance the treatment response rate of patients, and prolong their survival period. Taken together, it is anticipated that future combination treatment approaches will overcome resistance to ICIs and yield better survival outcomes for patients.

## Figures and Tables

**Figure 1 cancers-18-00039-f001:**
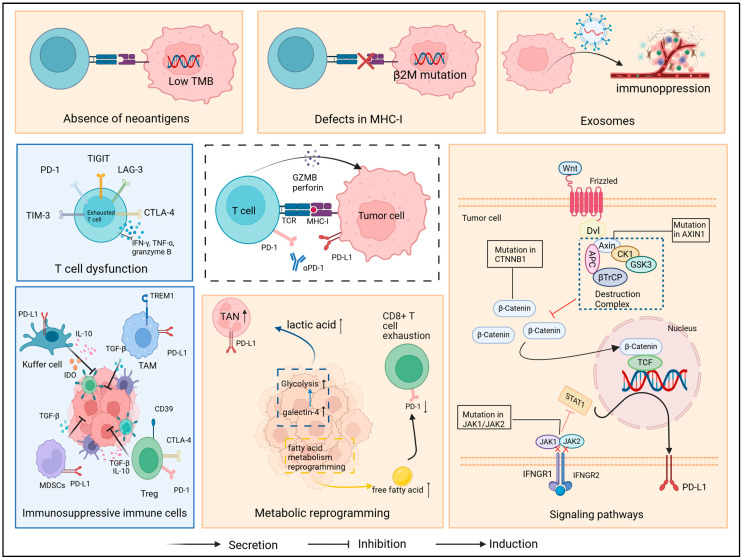
Mechanisms of primary resistance. The orange box represents internal factors within the tumor cell, while the blue box shows external factors outside the cell. Dashed lines indicate tumor cells that are sensitive to immunotherapy. Abbreviation: GZMB, granzyme B; TMB, tumor mutation burden; APC, *adenomatous polyposis coli*; IDO, indoleamine 2,3-dioxygenase; Dvl, disheveled; GSK3, glycogen synthase kinase 3; CK1, casein kinase 1; TCF, T cell factor; βTrCP, β-transducin repeat-containing protein; TME, tumor microenvironment. Created in https://BioRender.com, accessed on 21 December 2025.

**Figure 2 cancers-18-00039-f002:**
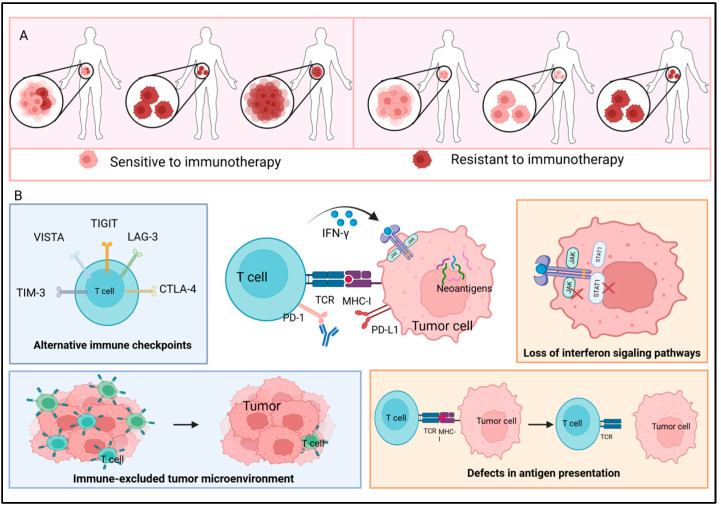
Acquired resistance (**A**) two modes of acquired resistance. (**B**) the mechanism of acquired resistance. The orange box represents internal factors within the cell, while the blue box shows external factors outside the cell. Acquired immune resistance to ICIs primarily manifests in two distinct forms: adaptive immune editing-related resistance and tumor-intrinsic evolution-driven resistance. In the former, tumors initially subjected to ICI-induced immune pressure gradually evade immune elimination through mechanisms such as clonal selection, downregulation of MHC-I expression, or impaired neoantigen presentation, ultimately developing a novel immune-tolerant phenotype, which is characterized by reduced neoantigen load, *B2M* mutations, MHC class I downregulation, and upregulation of immunosuppressive molecules (e.g., TIM-3, LAG-3). The latter form arises when tumor cells acquire novel biological properties via genetic mutations, signaling pathway reprogramming (e.g., JAK/STAT), or dedifferentiation, enabling them to sustain survival and escape immune surveillance despite immune attack. Created in https://BioRender.com, accessed on 21 December 2025.

**Figure 3 cancers-18-00039-f003:**
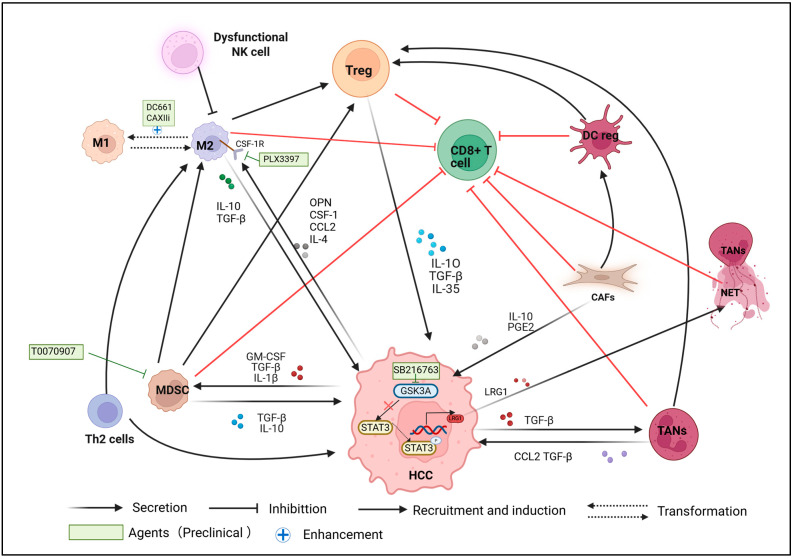
The immune microenvironment of HCC tumors and strategies for remodeling the immune microenvironment of tumors. Abbreviations: MDSC, myeloid-derived suppressor cell; LRG, leucine-rich α-2-glycoprotein 1; TANs, tumor-associated neutrophils; NET, neutrophil extracellular trap; TGF-β, transforming growth factor beta; PGE2, prostaglandin E 2; CSF-1, colony-stimulating factor-1; CSF-1R, colony-stimulating factor-1 receptor; GM-CSF, granulocyte–macrophage colony-stimulating factor; CAFs, cancer-associated fibroblasts; CAXIIi, carbonic anhydrases XII inhibitor. Created at https://BioRender.com, accessed on 21 December 2025.

**Table 3 cancers-18-00039-t003:** Potential therapeutic strategies in HCC.

Category	Agent	Target	Combination Therapies	Evidence Type	Trial Number	Ref.
Enhancing T cell-mediated immunity	GNOS-PV02	Personalized neoantigen-based vaccine	Pembrolizumab	Early clinical	NCT04251117	[171]
	NeoVAC	Personalized neoantigen vaccine	Anti-PD-1	Preclinical	NA	[172]
	AFP vaccine	Tumor antigen vaccine	Anti-PD-L1	Preclinical	NA	[170]
Targeting defects in antigen processing and presentation	Orlistat	FASN inhibitors	Anti-PD-L1	Preclinical	NA	[173]
	mDV-aCTLA-4 nano-system	Small nanovesicle derived from mature DCs	Anti-CTLA-4	Preclinical	NA	[174]
	Ilixadencel	Mature DC vaccine	Sorafenib	Early clinical	NCT01974661	[175]
	HCC tumor neoantigen-pulsed mature DC vaccine	Mature DC vaccine	Nivolumab	Early clinical	**NCT04912765**	NA
Targeting Wnt/β-catenin signaling pathway	siRNA	β-catenin	Anti-PD-1	Preclinical	Preclinical	[176]
	BC2059	β-catenin inhibitor	Cabozantinib	Early clinical	**NCT05797805**	NA
Targeting alternative ICIs	Cobolimab	Anti-TIM-3	Anti-PD-1	Early clinical	**NCT03680508**	NA
	BC3402	Anti-TIM-3	Tremelimumab + durvalumab	Early clinical	**NCT06608940**	NA
	Relatlimab	Anti-LAG-3	Nivolumab	Early clinical	**NCT04658147**	NA
	HLX53	Anti-TIGIT	Serplulimab + HLX04 (biosimilar to bevacizumab)	Early clinical	**NCT06349980**	NA
Targeting suppressive tumor microenvironment	CD39 knockout	Blocking CD39	Anti-PD-1	Preclinical	NA	[80]
	DC661	PPT1 inhibitor	Anti-PD-1	Preclinical	NA	[55]
	T0070907	MDSC	Anti-PD-L1	Preclinical	NA	[57]
	CAXII inhibitors	TAMs	Anti-PD-1	Preclinical	NA	[87]
	PLX3397	CSF1R inhibitor	Anti-PD-L1	Preclinical	NA	[54]
	Cabiralizumab	Anti-CSF1R	Anti-PD-1	Early clinical	**NCT04050462**	NA
	BMS-986253	Anti-IL-8	Anti-PD-1	Early clinical	**NCT04050462**	NA
	GT90001	Blocks TGF-β receptors	Nivolumab	Early clinical	**NCT05178043**	NA
	BMS-986205	IDO1 inhibitor	Nivolumab	Early clinical	**NCT03695250**	NA
	SAR439459	TGF-β inhibitor	Cemiplimab	Early clinical	**NCT04729725**	NA
	LY2157299	TGFβ receptor 1 inhibitor	Sorafenib	Early clinical	**NCT02178358**	NA

The bolded clinical trial numbers represent ongoing clinical trials, not completed, or whose results have not been officially published (the Ref. column is marked as NA). Abbreviations: FASN, fatty acid synthase; TIM-3, T cell immunoglobulin mucin-3; HCC, hepatocellular carcinoma; DC, dendritic cell; LAG-3, lymphocyte activation gene 3; TIGIT, T cell immune receptor with Ig and ITIM domains; PPT1, palmitoyl protein thioesterase 1; TGF-β, transforming growth factor-β; IDO1, indoleamine 2,3-dioxygenase 1.

## Data Availability

No new data were created or analyzed in this study. Data sharing is not applicable to this article.

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
