# Peer review of "Mechanisms of Immune Checkpoint Inhibitor Resistance in Hepatocellular Carcinoma and Strategies for Reversal"

_cancers, 2025, doi:10.3390/cancers18010039_

Round 1
Reviewer 1 Report
Comments and Suggestions for Authors
1. Some of the paragraphs are too long. It can be divided into short paragraphs according to the subtopics in the long ones to make reading and comprehension easier. One example is the paragraph on page 15.
2. Some of the abbreviations are not spelled out the first time they appear in the text and not in the abbreviation list at the back of the manuscript, for example, PVR, PPT1, etc.
3. It is better to sort the abbreviation list in alphabetical order to facilitate looking-up.
4. In line 56, “precise” is better replaced with “precision”.
5. In line 94, it is better to use “EGFR signaling” instead of “EGFR”.
6. In line 115, it is better to use “type I interferon” instead of “interferon-1 (IFN-1)”.
7. In line 122, Gal-9 is a ligand for TIM-3, not a receptor.
8. In line 203-204, “Neutrophils derived from” should be “PD-L1+ neutrophils induced by”.
9. In line 258, the “circ” part in “circTMEM181” may need to be described although it can be guessed as circulating.
10. In line 264, more explanation may be needed for the readers to understand why circCCAR1 can also promote resistance to anti-PD-1 therapy.
11. In line 307, “la-related” is usually written as “La-related”.
12. In line 311, the introduction of Riplet feels abrupt, more description of what this protein is may be needed.
13. In line 325, “ICI acquired resistance” should be “acquired ICI resistance”.
14. In line 359, “BTLAe” should be “BTLA”.
15. In lne 479, it is not clear whether it is necessary to or correct to abbreviate local treatment as LRT.
16. In the section of “4.3. Multiple ICIs”, the authors give a brief description of ipilimumab and sintilimab, but similar description is lacking for tremelimumab and durvalumab.
17. In line 643, it is difficult to understand “responses eventually progress”.
18. In line 691 to 692, it is better to add references for the promising strategies.
19. Figure 1 may lack the depiction for metabolic reprogramming described in the text.
20. It seems that Figure 2 is not mentioned or referred to in the text. Tumor heterogeneity is not in this figure. The antigen loss depicted in Figure 2B is not mentioned as a mechanism of acquired resistance in the text. Furthermore, no description is given in the text about the two modes of acquired resistance depicted in Figure 2A.
21. It appears that Table 1 is not referred to in the text.
22. Some of the hyphenations in words like “in-duce” (line 51) and “neu-trophils” (line 216) need to be removed.
Reviewer 2 Report
Comments and Suggestions for Authors
This was a well-organized comprehensive review based on extensive literature survey, including an array of preclinical and clinical cases. This manuscript was suggested to be published in this journal.
One discussion. What is the possibility for the application of CAR-T cells in the treatment of HCC?
One suggestion. It is better to give a brief introduction of the medicines in the treatment of HCC.
Reviewer 3 Report
Comments and Suggestions for Authors
Review Report
concerning manuscript no. cancers-4007102
entitled „Mechanisms of immune checkpoint inhibitor resistance in hepatocellular carcinoma and strategies for reversal”
Brief Summary: The authors present a comprehensive review of the mechanisms of primary and acquired resistance to checkpoint inhibitors in hepatocellular carcinoma (HCC), including: (i) lack of neoantigens and impaired antigen presentation, (ii) activation of signaling pathways (Wnt/β-catenin, JAK/STAT, STAT3), (iii) T cell and NK cell dysfunction, (iv) the role of immunosuppressive cells in the tumor microenvironment, exosomes, and metabolic reprogramming, (v) acquired resistance (antigen loss, defects in IFN-γ signaling, alternative checkpoints), and (vi) current and emerging therapeutic strategies (combination therapies, novel targets, neoantigen vaccines, TME modulation). The manuscript is highly timely, encompassing work from 2023–2025 and encompassing numerous data from phase II–III trials. Overall: This is a valuable, up-to-date review with the potential to be frequently cited. However, it requires significant substantive and editorial revisions to enhance clarity, balance the proportions between sections, and improve the coherence of detail. Recommendation: Major revision.
Major comments:
- In several places, resistance mechanisms are described primarily based on studies in melanoma or lung cancer (e.g., B2M/antigen presentation in section 2.2.1, JAK1/2/IFN-γ mutations in 2.2.4), with a brief statement that "data in HCC are lacking." I suggest adding an introductory sentence in section 2.2 to clearly indicate when the authors are referring to "proof-of-principle" from other cancers and that in HCC, these are hypotheses for now. I also suggest creating a short table that, for each resistance mechanism, provides the type of resistance (primary/acquired), the data source (HCC – clinical / HCC – preclinical / other cancers), and the potential "clinical readiness" (e.g., existing drugs, ongoing trials). This will help the reader assess to what extent the data are already established in HCC and to what extent they are hypothetical.
- Section 2.1 (primary resistance) is very extensive and detailed (neoantigens, MHC, TME, suppressor cells, metabolism, etc.), while section 2.2 (acquired resistance) is relatively short and more general. This dissonance creates a negative impression. Please therefore expand section 2.2 with the following information: (i) more specific examples of acquired resistance in HCC (even individual case reports, if available), (ii) a brief summary of how often (estimated) acquired resistance is observed in HCC clinical trials (e.g., progression rate after initial response).
- Section 3 (biochemical predictors) focuses on PD-L1, TMB/MSI, and the microbiota, while some very interesting biomarker candidates (TGF-β, AFP, PIVKA-II, inflammatory parameters, ALBI) appear only in the conclusion. Please expand section 3 with an additional subsection titled, for example, "Other emerging biomarkers," which would include changes in AFP/PIVKA-II after 6 weeks of therapy, inflammatory markers (NLR, PLR, CRP, CRAFITY), and liver function parameters (ALBI). For PD-L1, it is worth adding a brief explanation of the differences between TPS, CPS, and IC scores, as well as the information that different thresholds and antibody clones were used in HCC studies, which complicates comparison of results and may partially explain the contradictory observations.
- In the chapter on TMB/MSI, please add a comment that despite the low frequency of TMB-H/MSI-H in HCC, other genomic features (e.g. CTNNB1 mutations, angiogenic profile, inflammatory signatures) may have a higher predictive value – which also links the text with the earlier description of the Wnt/β-catenin pathway.
- Sections 4 (“Combined therapy to overcome ICIs resistance”) and 5 (“Potential therapeutic strategies”) partially overlap: both discuss combinations of ICIs with TKIs, TME modulation, alternative checkpoint blockade, etc. I suggest focusing in Section 4 only on currently used or clinically advanced strategies (atezo+bev, durva+tremeli, camrelizumab+rivoceranib, ICI+TACE/RFA/Y-90, RT+ICI, etc.), with an emphasis on ORR, PFS, OS, and registration status. In section 5, I suggest describing only prospective/preclinical strategies, with a clear division by mechanism: (i) enhancing the T-cell response (neoantigen vaccines, AFP, oncolytic viruses), (ii) targeting the TME (TAMs, MDSCs, etc.), (iii) modulation of metabolism (FASN, TACC3, Riplet), (iv) novel checkpoints (TIGIT, LAG-3, etc.).
- Figures 1–3 summarize the mechanisms well, but the references to them in the text are rather terse. In sections 2.1 and 2.2, please add 1 – 2 sentences of "guidelines" after Figures 1 and 2 (e.g., "As shown in Figure 1, intracellular mechanisms include X, Y, Z, while extracellular mechanisms include A, B, C"). I believe that in Figure 3, consider adding simple icons/labels for therapies already in clinical trials vs. early preclinical trials – this will increase the practical value of the illustration.
- Table 1 – Combinations with ICIs, is very valuable but requires minor refinements: (i) Please add a column titled "Line of therapy / setting" (1st line, 2nd line, post-atezo+bev, etc.); (ii) Please indicate which studies are completed with published results and which are ongoing (e.g., by bolding the trial number or adding a footnote).
- Table 2 – potential strategies: (i) I suggest adding a column “Type of evidence (preclinical / early clinical / phase II+)” instead of describing it only in the “Phase” column (ii) pls check for abbreviations and typos.
Minor comments
- Please check that all abbreviations are correct and explained where they are first used.
- Please check the spelling of drug names and other substances, e.g., "Carrizumab" (Table 1) – this most likely refers to Camrelizumab.
- Please check and correct typos (e.g., "fatty acid synthase" → "fatty acid synthase").
- Please standardize the spelling of abbreviations, e.g., STAT 3 vs. STAT3, VEGFα vs. VEGF-A, etc.
- The English language is generally good, but I would recommend editing by a native speaker to improve flow and shorten some very long sentences (especially in sections 2.1.5–2.1.8 and 5.5).
- 1–2 sentences may be added to the abstract to clearly highlight what is new in this review compared to existing ones (e.g. integration of the latest metabolic and TME data).
- When citing Phase III trials (IMbrave150, HIMALAYA, CARES-310, CheckMate 9DW), it would be helpful to briefly state the main endpoints (ORR, PFS, OS) in the text, even if the details are in the table.
- In several places in the manuscript, a single connecting sentence would be helpful, for example, between the descriptions of NETs, MDSCs, and macrophages – the paragraphs currently move very quickly from one cell type to another, which can be difficult for a reader less familiar with the topic.
- References [150] and [151] describe the same article (Kelley et al. 2021).
- Please add the DOI numbers to all references. This will make it easier to find the relevant cited publication if a specific point interests the reader.
Summary: The manuscript is substantively strong, up-to-date, and well-documented, and the topic is clinically relevant. My main concerns include the structure of the therapeutic sections, the balanced description of acquired resistance, the clear separation of data for HCC versus other cancers, and the refinement of tables/figures and editorial details. With the suggested changes, the article has the potential to become a comprehensive, reference review of ICI resistance in HCC.
Comments on the Quality of English Language- The English language is generally good, but I would recommend editing by a native speaker to improve flow and shorten some very long sentences (especially in sections 2.1.5–2.1.8 and 5.5).
Reviewer 4 Report
Comments and Suggestions for Authors
Respected Xin-ye Dai with coauthors have presented a very thorough and comprehensive review dedicated to the mechanisms of resistance to immune checkpoint inhibitors in hepatocellular carcinoma. The relevance of new developments in cancer therapy is beyond doubt, and immunotherapy remains a highly promising direction. Immune checkpoint inhibitors (ICIs) have changed the therapeutic paradigm for many cancers, significantly improving survival for patients who previously had incurable disease. However, their effectiveness varies, and not all patients achieve durable responses, necessitating a deep understanding of the mechanisms of action and resistance to optimize therapy. Therefore, I consider this review urgent and valuable from both a fundamental and practical perspective.
The review is clearly structured, beginning with an introduction and ending with conclusion. The main body consists of four major sections. It starts with the fundamental basis of resistance to ICIs, followed by discussion of biochemical predictors of response, and then proposes combination therapy strategies to overcome resistance.
No similar comprehensive review has been published in the past decade or earlier. This review impresses with its depth and extensive information. It includes 184 references, more than half published in the last five years, thus presenting the most up-to-date data.
The authors' conclusions are correct and supported by references. The review contains two informative tables summarizing key approaches to overcoming resistance, as well as three original figures illustrating resistance mechanisms and strategies for remodeling the immune microenvironment.
I highly appreciate the authors' work and believe it deserves publication. However, I have a few suggestions for improving the presentation:
- In the introduction, after the sentence "have demonstrated significant efficacy in numerous malignancies and have also been extensively applied in the treatment of HCC" (line 36), I would suggest adding general information about the classes of immune checkpoint inhibitors. For example: The main classes of checkpoint inhibitors include CTLA-4 inhibitors on T cells (Ipilimumab, Tremelimumab), PD-1 inhibitors on activated T cells (Nivolumab, Pembrolizumab, etc.), PD-L1 inhibitors on tumor cells (Atezolizumab, Durvalumab, Avelumab), and the LAG-3 inhibitor on T cells (Relatlimab, in fixed combination with PD-1 blocker Nivolumab). Among these, Nivolumab and Atezolizumab in combination with VEGF blockade are used for hepatocellular carcinoma treatment. Or something like that. This phrase will provide a logical link to the subsequent text.
- I propose a slight revision of Figure 1. Currently, the figure suggests a logical progression that does not match the order of the text: the first section is "Absence of neoantigens," followed by "Defects in MHC-I." If the authors decide to keep the figure at the start of the chapter, I would ask to reorganize it to reflect the logical order of the text and provide brief captions for each panel. Alternatively, moving it to the end of section 2, as done with Figures 2 and 3, would avoid the need for captions and reformatting. In that case, please add a note about the white panel in the center of the figure, as only the orange and blue panels are mentioned in the caption. Also, please correct the typo "Exsomes" on the figure.
- For section 3.2 title, I recommend avoiding abbreviations (unless chemical names) in headers.
- The conclusion is currently written in the format of a discussion, containing references and problem statements, which is inappropriate. I suggest splitting the "Conclusion" section into two: "Discussion" (lines 640–683) and "Conclusion" (lines 684–701). References [5] and [184] and, as well as the mention of Table 2, should be removed from the "Conclusion" section.
Round 2
Reviewer 3 Report
Comments and Suggestions for Authors
Thank you very much for addressing my comments and suggestions. I appreciate the careful and thorough revisions you have made to the manuscript. All concerns have been satisfactorily resolved, and the revised version meets the required standards.
I am pleased to confirm that the manuscript is suitable for publication.